# Polarized Object Surface Reconstruction Algorithm Based on RU-GAN Network

**DOI:** 10.3390/s23073638

**Published:** 2023-03-31

**Authors:** Xu Yang, Cai Cheng, Jin Duan, You-Fei Hao, Yong Zhu, Hao Zhang

**Affiliations:** 1College of Electronic Information Engineering, Changchun University of Science and Technology, Changchun 130022, China; 2College of Computer Science and Technology, Changchun University of Science and Technology, Changchun 130022, China

**Keywords:** three-dimensional reconstruction, polarization images, surface normal vector, specular reflection model, reflective region localization

## Abstract

There are six possible solutions for the surface normal vectors obtained from polarization information during 3D reconstruction. To resolve the ambiguity of surface normal vectors, scholars have introduced additional information, such as shading information. However, this makes the 3D reconstruction task too burdensome. Therefore, in order to make the 3D reconstruction more generally applicable, this paper proposes a complete framework to reconstruct the surface of an object using only polarized images. To solve the ambiguity problem of surface normal vectors, a jump-compensated U-shaped generative adversarial network (RU-Gan) based on jump compensation is designed for fusing six surface normal vectors. Among them, jump compensation is proposed in the encoder and decoder parts, and the content loss function is reconstructed, among other approaches. For the problem that the reflective region of the original image will cause the estimated normal vector to deviate from the true normal vector, a specular reflection model is proposed to optimize the dataset, thus reducing the reflective region. Experiments show that the estimated normal vector obtained in this paper improves the accuracy by about 20° compared with the previous conventional work, and improves the accuracy by about 1.5° compared with the recent neural network model, which means the neural network model proposed in this paper is more suitable for the normal vector estimation task. Furthermore, the object surface reconstruction framework proposed in this paper has the characteristics of simple implementation conditions and high accuracy of reconstructed texture.

## 1. Introduction

As a key research object in the field of machine vision and imaging, 3D reconstruction technology has a wide range of utility. Currently, 3D reconstruction techniques have been applied to virtual battlefields, unmanned vehicles, deep space exploration, medical imaging, scene reconstruction, and other fields [1]. Studies have shown that polarization imaging technology can better highlight the surface texture information of objects than intensity images [2], and thus a method to achieve 3D reconstruction using polarization technology gradually comes within the vision of researchers.

From a practical point of view, studies on 3D reconstruction focus on how to reconstruct 3D structures from 2D images, due to the broader applicability of this direction. Under ideal conditions, too many objective conditions exist for the study, which makes the task of achieving 3D reconstruction too onerous; for non-ideal scenes, some of the conditions are not achievable, which means that 3D reconstruction cannot be realized. Thus, a method to recover 3D structures from 2D images is the focus of this paper, i.e., the estimation of height information in order to achieve reconstruction [3]. Since the relationship between the object surface and the normal vector is unknown, the reconstruction of the object surface from the intensity image is very unsatisfactory. As scholars continue to study the relationship between the direction of vibration and the direction of the propagation of light waves, they found that the relationship between these two can be obtained by the properties of light (polarization). Koshikawa et al. [4] were the first to extract polarization information from reflected light and then reconstruct the smooth surface of the object. Later, B. Wolff used polarized images combined with stereo vision techniques to construct the surface normal vector information of an object [5]. In addition, Wolff also used Fresnel’s formula to explain the polarization characteristics of reflected light, and classified (including specular reflected light, diffuse reflected light, and diffracted light) and described the reflected light of the object, and clarified the difference between different types of reflected light [6]. The results of this study laid the theoretical foundation for the subsequent analysis of the polarization state of reflected light from the target surface. Since the conditions of the surface generating diffracted light reflection are very special, they can be ignored in polarized 3D imaging studies, and the current research mainly focuses on the use of specular reflected light and diffuse reflected light polarization properties to achieve the reconstruction of the target 3D profile. However, the physical model provided by Wolff for normal vector solving has four feasible solutions under the specular reflection model and two feasible solutions under the diffuse reflection model, so researchers have started to explore how to resolve the ambiguity, i.e., to obtain the true normal vector from six feasible solutions.

Gary A. et al. [7] found that there is a polarization effect on partially diffuse reflected light and that the mapping of the polarization of diffuse reflected light to the incident angle is unique, so they proposed to use the polarization from diffuse reflected light to calculate the uniquely determined incident angle. Smith et al. [8] used polarization and shadow information constraints as linear equations when solving for height information. Mahmoud et al. [9] used texture information from shadows. Miyazaki et al. [10] transformed the RGB-measured color information into S-space and used the brightness, hue, and saturation defined in S-space to construct direct correspondence with the target color, diffuse reflection, and highlight reflection parameters [11] to separate the specular and diffuse reflected light from the target surface, solving the problem of inaccurate DOP (degree of polarization) solutions in the presence of both specular and diffuse reflections. However, the above methods can only partially solve the normal vector ambiguity, and the experimental results have the problems of large errors and poor robustness. For this reason, Zhang et al. [12,13] proposed to further resolve the normal vector ambiguity by combining parallax angle and point cloud data to estimate the normal vector, combined with the Poisson distribution to achieve surface reconstruction; the accuracy of its normal vector estimation is low, but the reconstruction results have some significance. With the rapid development of deep learning, Ba et al. [14] first input polarization information as a priori knowledge into the U-net network for normal vector estimation, and the results showed that the normal vector accuracy obtained by Ba et al. was higher. Chen et al. [15] proposed a pre-trained network, and further pointed out in the paper that the polarization surface normal vector [5] combined with polarization map as input could obtain a higher accuracy estimated normal vector.

In summary, the ambiguity generated by polarized images has not a good solution. Although neural networks have a strong learning ability, this method is still not widely used to solve the ambiguity of surface normal vectors. Only U-net networks [14,15] have made some progress in resolving normal vector ambiguity, while Gan networks have been little studied in this area. The essence of how convolutional neural networks [14,15] solve normal vector ambiguity is image fusion, and studies have shown that generative adversarial networks [16] have demonstrated good results in the processing of infrared images, polarized images, and intensity images. The object under study in this paper needs to distinguish the target from the background, i.e., it requires a certain semantic segmentation capability. Therefore, we fuse the generative adversarial network architecture with the U-net network architecture, supplemented by Resnet for the normal vector estimation task, and further demonstrate that the generative adversarial network can be better applied to the normal vector estimation tasks. Based on this paper, a holistic framework for object surface reconstruction from single visual images is proposed.

The work in this paper can be summarized as follows:A set of frameworks is proposed to reconstruct the surface of an object from polarized images only, as shown in Figure 1. First, six feasible solutions for zenith and azimuth angles are obtained using the physical model, and six polarized surface normal vector maps are reconstructed. Secondly, the RU-GAN model proposed in this paper is used for the normal vector estimation task, and then the height map of the object surface is obtained from the energy function. Finally, the object surface is constructed from the height map.The use of the spectral properties under the specular reflection model is proposed to achieve reflective localization on the surface of an object. Further, data preprocessing using bilinear interpolation optimizes the neural network model input and narrows the range of reflective region in the image.The uncertainty of the zenith angle and azimuth angle is solved using a neural network. Six polarized surface normal vectors are introduced into the RU-Gan network to realize the real normal vector estimation task. Since the six polarized surface normal vectors are enriched with six feasible solutions for zenith angle and azimuth angle, so the estimated normal vector obtained by using them as network inputs is more accurate. The RU-Gan network proposed in this paper uses jump compensation in the generator part, which greatly reduces the information loss during the learning process. The content loss functions reconstructed in this paper include cosine loss and mean squared loss, which enhance the ability of the network to learn texture information and numerical fitting information, respectively.

## 2. Fundamentals

Since the six polarized surface normal vector maps are richer in numerical fitting information than the polarization and intensity maps [14,15], this section introduces the origin and correction of the six polarized surface normal vectors, namely, the principle of the polar coordinate of normal vectors is firstly stated. Therefore, it is shown that the normal vector diagram is composed of the azimuthal angle φ and zenith angle θ. Secondly, the physical model proposed by Wolff [5] was used to solve the azimuthal angle φ, zenith angle θ, and the six polarization surface normal vectors were reconstructed with these two angles. Finally, the six polarized surface normal vectors are corrected for the reflective regions using the specular reflection model.

### 2.1. Polar Coordinate Representation of Surface Normal Vector

For any surface normal vector *N*, its pointing can be expressed in right-angle coordinates (x,y,z). The angle between the projection of the normal vector *N* onto XOY and the positive direction of the *X*-axis is the azimuth angle φ. The angle between the normal vector *N* and the *Z*-axis is the zenith angle θ, as shown in the following figure.

The three channels of the normal vector map store the *X*, *Y*, and *Z* axis coordinates of the normal vector, so the essence of the normal vector map is the spatial vector of the point. As can be seen in Figure 2, the relationship between the right angle coordinates of a point in space and polar coordinates, so as long as the azimuth angle *φ* and zenith angle *θ* are known, can further reconstruct the normal vector map, and the relationship equation between the two is as follows.
(1)$$\left\{\begin{array}{c} X=\sin(\varphi )\sin(\theta )\\ Y=\cos(\varphi )\sin(\theta )\\ Z=\cos(\theta ) \end{array}\right.$$

### 2.2. Light Reflection Model

As shown in Figure 3, incident light will be reflected and refracted in the interior or surface of different materials, which in turn are classified as diffuse and specular reflections. From this, Wolff assumes two cases based on the Fresnel equation and optical geometry relations [5], the pure specular reflection model and pure diffuse reflection model, and solved the azimuthal angle φ and zenith angle θ according to these two models. 

The relationship between the DOP $P_{\mathrm{spec}}$ and zenith angle θ, phase angle ϕ, and azimuth angle $\varphi_{S}$ under the pure specular reflection model, is as follows.
(2)$$\left\{\begin{array}{c} P_{spec}=\frac{2{sin}^{2}\theta cos\theta \sqrt{n^{2}\;-\;{sin}^{2}\theta }}{n^{2}\;-\;{sin}^{2}\theta \;-\;n^{2}{sin}^{2}\theta \;+\;2{sin}^{4}\theta }\\ \varphi_{s}=\phi +\frac{\pi }{2}or\;\phi +\frac{3\pi }{2} \end{array}\right.$$

The relationship between the DOP $P_{d}$ and the zenith angle θ, phase angle ϕ, and azimuth angle $\varphi_{d}$ under the diffuse reflection model, is as follows.
(3)$$\left\{\begin{array}{c} P_{d}=\frac{{\left(n\;-\;\frac{1}{n}\right)}^{2}{sin}^{2}\theta }{2\;+\;2n^{2}\;-\;{\left(n\;+\;\frac{1}{n}\right)}^{2}{sin}^{2}\theta \;+\;4cos\theta \sqrt{n^{2}\;-\;{sin}^{2}\theta }}\\ \varphi_{d}=\phi \;or\;\phi +\pi \end{array}\right.$$

In Equations (2) and (3), the zenith angle cannot be obtained by DOP inversion, so this paper will solve it by the dichotomous method. In Equation (2), one polarization value will produce two zenith angle solutions $\theta_{1}/\theta_{2}$; in Equation (3), one polarization value will produce one zenith angle solution $\theta_{1}$, as shown in Figure 4.

Where the solution equations [17] for the DOP P and phase angle *φ* are as follows.
(4)$$\left\{\begin{array}{c} P=\frac{\sqrt{S_{1}^{2}\;+\;S_{2}^{2}}}{S_{0}}=\frac{\sqrt{{\left(I_{0}\;-\;I_{90}\right)}^{2}\;+\;{\left(I_{45}\;-\;I_{135}\right)}^{2}}}{\frac{1}{2}\left(I_{0}\;+\;I_{45}\;+\;I_{90}\;+\;I_{135}\right)}\\ \varphi =\frac{1}{2}\arctan\left(S_{2},S_{1}\right)=\frac{1}{2}{tan}^{-1}\frac{I_{45}\;-\;I_{135}}{I_{0}\;-\;I_{90}} \end{array}\right.$$

In Equation (4), $I_{0}$, $I_{45}$, $I_{90}$, and $I_{135}$ are polarization images in the directions of 0°, 45°, 90°, and 135°, respectively.

### 2.3. Reflective Area Correction

Since the real normal vector map does not contain reflective regions, and most of the images obtained by the camera contain reflective regions, such regions not only cause the estimated normal vectors to deviate from the true normal vectors, but also cause the six polarized surface normal vectors obtained by Equations (2) and (3) to contain reflective regions as well. Traditional reflective region localization requires manual setting of thresholds and is less well achieved [18]. In this paper, we propose to use the spectral properties of specular reflection model to obtain the reflective region map $I_{L}$, as in the following equation.
(5)$$I_{L}(x,y)=\sum_{x,y\in I}I_{g>b}(x,y)$$
where *I* is the normal vector map composed when θ is smaller than the Brewster angle and $\varphi_{S}=\phi +\frac{\pi }{2}$ in Equation (2). *g* and *b* represent the second and third channels of the image *I*, respectively, and bilinear interpolation [19] of reflective points (x,y) for I is carried out based on the reflective region map *I_L_*, as follows.
(6)$$I_{af}(x,y)=\frac{1}{n}\sum_{i\in x,y}^{n=2}\frac{I_{i}-Min_{i}}{Max_{i}-Min_{i}}(I\_Max_{i}-I\_Min_{i})+I\_Min_{i}$$
where, $I_{af}$ is the image after interpolation, i is the direction of interpolation, *n* is the number of interpolation directions (here it is 2). $I\_Max_{i}$ and $I\_Min_{i}$ are the maximum and minimum pixel values in the interpolation direction, $Max_{i}$ and $Min_{i}$ are the maximum and minimum coordinate values in the interpolation direction, and $I_{i}$ is the coordinate of the interpolation point.

A schematic of the partially polarized surface normal vector reflective region is shown in Figure 5, while all six polarized surface normal vectors are actually subject to correction. As in Figure 6, the six polarized surface normal vectors corrected by the reflective region are input to the neural network model as *IN*1, and 0°, 45°, and 90° direction polarized images as *IN*2.

## 3. Proposed Method

Inspired by the literature [17], this paper proposes a jump-compensated U-shaped generative adversarial network (RU-Gan) based on jump compensation to solve the normal vector ambiguity problem, i.e., to perform the true normal vector estimation task, where the generator is shown in Figure 6. To ensure that the network can have the correct learning direction during adversarial learning, a content loss function will be introduced while using the true normal vector to supervise the estimation of the normal vector. The process of getting height maps from normal vectors mostly requires the assistance of point cloud data, and to solve this problem, this paper constructs an energy function starting from the association between normal vectors and gradients. In other words, the output estimated normal vector is finally combined with the energy function to solve the height map and then reconstruct the object surface.

### 3.1. Neural Network Model

In this paper, we propose to input the polarized surface normal vectors as a priori knowledge into the Gan network, and thus resolve the normal vector ambiguity while obtaining the estimated normal vectors. The reconstructed generator is divided into three modules, namely, encoder module (orange), decoder module (dark green), and output module (blue). The size of the convolution kernel of the output module is 3 × 3, and the number of input and output channels are 32 and 3, respectively. The overall network architecture of the generator is as follows.

The network structure of U-Net can be divided into two parts, downsampling and upsampling; the downsampling part decomposes the image into feature maps with different levels of abstraction, while the upsampling part restores the feature maps back to the original image size. In the central part of the network, the U-Net also includes some jump connections that combine the low-level and high-level features extracted during downsampling and upsampling, so using the U-net network as a generator can yield a better estimated normal vector map. As shown in Figure 6, the role of the generator is to realize the mapping from multiple images to the real normal vector image, and the specific details of the encoder module and decoder module are shown in Figure 7 and Figure 8. The three-channel image *IN2* of the network input consists of grayscale polarized images in the 0, 45, and 90 directions, and *IN1* is the six normal vector maps consisting of azimuth and zenith angles solved by Equations (2) and (3) and corrected by reflective regions, and the output of the network is the estimated normal vector $\hat{N}$ as follows
(7)$$\hat{N}=f\left(IN1,IN2\right)$$
where f is the network model. The original U-net network loses some color and texture information during encoding and decoding, so this paper adds the Resnet network to the encoder (Figure 7) and decoder (Figure 8) parts to compensate for the problem. Inputting polarization image *IN*2 and polarization surface normal vector image *IN*1 for the network will enhance the ability of the model to learn semantic information and numerical fitting information. The normalized *X*, *Y*, and *Z* channels should take values between −1 and 1, so the LeakyReLU activation function can achieve the purpose of retaining the negative information better.

As shown in Figure 7 and Figure 8, the specific expansion on the left side is shown on the right side of the middle sign of the figure. Since the polarization image is richer in texture information than the intensity image [15], this paper proposes to reintroduce the semantic information provided by polarized images in the process of decoding, to enhance the surface features of the estimated normal vectors. The size of the convolutional kernels used in both the encoder and decoder modules is 3 × 3, and both the native U-net jump connection and the introduction of the polarized image are implemented in the Concat position of the decoder. In the process of testing the network performance, this paper finds that the InstanceNorm normalization layer can better maintain the original texture information of the object, which is consistent with the results reported in [15], and different from [14]. D1 and U1 in the module are the number of input channels, and D2 and U2 are the number of output channels. The sizes of D1:D2 in the encoder module are 21:32, 32:64, 64:128, 128:256, and 256:512, and the sizes of U1:U2 in the decoder module are 1280:512, 896:256, 448:128, 224:64, and 117:32, respectively.

The task of the discriminator is to distinguish the real reliability of the generated image by the generator, so this paper will perform the feature number reduction operation on the generated image and the real image. The model learns the texture best when the Stride is 2, but the color distortion problem occurs, while the model learns the color and texture information most closely when the Stride is 4, and the accuracy of the estimated normal vectors is the highest, so the Stride is set to 4 in the discriminator. The output is normalized to between 0 and 1 using the sigmoid function, which represents the true probability of each pixel point estimate, and the discriminator network architecture is shown in Figure 9. 

### 3.2. Loss Function and Training Parameters

The loss function is divided into two components, namely, generator loss $L_{G}$ and discriminator loss $L_{D}$. The generator loss function $L_{G}$ consists of two components, adversarial loss $L_{\mathrm{adv}}$ and content loss $L_{c}$. In order to obtain the probability distribution of the true normal vector map $I^{\mathrm{ft}}$, the adversarial loss of *G* is defined as
(8)$$L_{\mathrm{adv}}=E\left(\mathrm{lg}\left(1-D\left(I^{f}\right)\right)\right)$$
where *E* is the expectation; $D\left(I^{f}\right)$ is the value of the fused image $I^{f}$ generated by the generator *G* obtained by the discriminator *D*; $L_{\mathrm{adv}}$ is the loss generated by the image $I^{f}$ generated by *G* saved in $I^{\mathrm{ft}}$. Content loss $L_{c}$ includes cosine loss $L_{cos}$ and mean square error loss $L_{\mathrm{mse}}$. The cosine loss refers to the difference in pointing between the estimated normal vector and the true normal vector, as follows.
(9)$$L_{cos}=1-\frac{A\cdot B}{\left\|A\|B\right\|}=1-\frac{\sum_{i=1}^{n}A_{i}\times B_{i}}{\sqrt{\sum_{i=1}^{n}{\left(A_{i}\right)}^{2}}\times \sqrt{\sum_{i=1}^{n}{\left(B_{i}\right)}^{2}}}$$
where *n* is 3, the closer the $L_{cos}$ value is to 0, the closer the angle between the vectors is to 0°, i.e., the more similar the two vectors are. When the angle between the two vectors is 90°, the $L_{cos}$ value is 1. When the two vectors are in opposite directions, the $L_{cos}$ value is 2. The mean square error loss refers to the difference between the estimated normal vector and the true normal vector values, which is based on the following principle.
(10)$$L_{\mathrm{mse}}=E\left[\frac{1}{mnk}\sum_{c=1}^{k}\sum_{i=1}^{m}\sum_{j=1}^{n}{\left(I_{i,j,c}^{\mathrm{ft}}-I_{i,j,c}^{f}\right)}^{2}\right]$$
where *k* is the number of channels and *k* = 3. In summary, the generator loss function $L_{G}$ expression is
(11)$$L_{G}=L_{\mathrm{adv}}+\lambda \left(L_{cos}+L_{\mathrm{mse}}\right)$$

The discriminator *D* is used to discriminate between the generated image $I^{f}$ and the real image $I^{\mathrm{ft}}$, the output of *D* is a value from 0 to 1, which represents the probability of the input image, so the adversarial loss of *D* is
(12)$$L_{D}=E\left[-\mathrm{lg}\left(1-D\left(I^{f}\right)\right)\right]+E\left[-\mathrm{lg}\left(D\left(I^{\mathrm{ft}}\right)\right)\right]$$

In this paper, the estimation normal vector algorithm is trained and tested using PyTorch and NVIDI Tesla T4 16G, and Adam is chosen as the optimizer. During training, the optimal learning rate is sought by setting the initial learning rate to 0.01 and decaying by ten percent every 20 rounds, and the final learning rate is set to 0.001. The size of the network input after resizing and concatenation is 512 × 512 × 21 and the dataset is used as DEEPSFP dataset [14].

### 3.3. Object Surface Reconstruction

Since the direction of the normal vector of any surface is perpendicular to that surface, the relationship between the normal vector and the gradient of the surface can be known. That is, assuming that the two gradients on the surface of the object are *U* and *V*, then the normal vector should be perpendicular to both gradients, i.e., the vectors are multiplied to 0. The gradients in the *U* and *V* directions can be solved by the height map; inspired by the literature [20], this paper solves the height map by constructing an energy function between the surface normal vector and the surface gradient, as follows.
(13)$$E=\sum_{u,v}{\left(t_{u,v}^{\|}\cdot n_{u,v}\right)}^{2}+{\left(t_{u,v}^{⊥}\cdot n_{u,v}\right)}^{2}$$
where the expressions for the gradients in the *U*-direction and *V*-direction are as follows.
(14)$$\left\{\begin{array}{c} t_{uv}^{\|}=[1,0,Z(u+1,v)-Z(u,v)]\\ t_{uv}^{⊥}=[0,1,Z(u,v+1)-Z(u,v)] \end{array}\right.$$
where *Z* is the height map to be found. Ideally, the value of the energy function should be zero, so minimizing it will give a more accurate height map. In this paper, the method of gradient descent [21] is used to solve the height information map.

## 4. Experimental Results and Analysis

### 4.1. Experimental Results and Analysis

#### 4.1.1. Experimental Validation of Resnet

Since the native U-Net network is too deep, it generates problems such as the loss of a large amount of texture information in the encoder and decoder parts, gradient dissipation and degradation, and excessive errors. Therefore, in the initial design of the generator, this paper adds a jump connection between every two convolutional layers to strengthen the connection between adjacent convolutional layers. In this section, three comparative experiments are performed to demonstrate the advantages of Resnet: (a) removing Resnet from the encoder part, i.e., increasing the information loss of the encoder; (b) removing Resnet from the decoder part, i.e., reducing the learning capability of the decoder; and (c) using the complete network architecture of RU-Gan in this paper to generate fused images. All these comparison experiments are performed under the same setup and the fusion results are shown in Figure 10.

It can be seen through the red box that the first column leads to a decrease in the model’s ability to learn the texture because of the lack of Resnet in the encoder part; the second column shows that the model learns less information about the true normal vector color during the last upsampling, and this leads to an overall fading of the estimated normal vectors obtained. It can be seen that the use of Resnet structure in the downsampling process can increase the ability of the model to learn texture to some extent, and the addition of the Resnet structure in the upsampling process can correctly guide the model to learn the fitting information of the normal vectors.

#### 4.1.2. Experimental Verification of Content Loss

The content loss of the generator is divided into two parts, i.e., cosine loss and mean squared loss. To verify the effectiveness of the proposed content loss, three comparison experiments are designed in this paper: (a) Removing the cosine loss function, which implies a reduced ability to learn normal vector pointing when the network model is learned. (b) Removing the mean squared loss function, which will result in the lack of numerical fit information to guide the network learning. (c) Using the complete content loss guiding model learning in this paper. All these comparison experiments were performed under the same setup and the fusion results are shown in Figure 11.

As shown in Figure 11, the first column has better overall learning than the second column. However, it can be seen through the red and yellow box that the second column has better detail learning, specifically on individual pixel points, such as the ears and hind legs area of the beast. This is because the first and second columns are missing the normal vector pointing and numerical fitting guidance, respectively, which leads to the model learning deviating from the correct direction. In summary, using cosine loss and mean squared loss together as content loss can play a complementary and co-facilitating role.

### 4.2. Analysis of Results

MAE (mean absolute error) is a common evaluation index in normal vector estimation task; the higher MAE means that the overall result is different from the true value. To better demonstrate the advanced nature of the proposed network model, six representative target objects will be taken for metric analysis, as shown in Table 1. For better data visualization, the comparison graph of the algorithms for estimating normal vectors will be cropped around the edges. In this paper, the metrics are first compared with the estimated normal vectors obtained from the traditional physical model, as shown in Figure 12, from left to right: the real normal vectors, the algorithm proposed in this paper, the algorithm of Zhang [12] or Yang et al. [13], the algorithm of Mahmoud et al. [9], and the algorithm of Miyazaki et al. [10].

In Figure 12, it can be seen that the algorithm proposed in this paper obtains a higher accuracy of estimated normal vectors than the other three algorithms. In terms of color, the estimated normal vector results in this paper are more closely matched to the true normal vector. The estimated normal vectors obtained by algorithms [12,13] have unnatural red–blue junction excess, which is due to Yang and Zhang et al. ignoring the actual meaning of the normal vectors when performing data processing; while the estimated normal vectors obtained by algorithms [9,10] are substantially distorted from the true normal vectors because these two algorithms can only obtain rough azimuths. As shown in Table 1, the estimated normal vectors obtained in this paper improve the accuracy by about 7° compared with Zhang and Yang et al., and about 25° and 30° compared with Mahmoud and Miyazaki et al., respectively. It can be seen that the normal vector estimation task using a convolutional neural network has certain advantages. At present, there are few studies on inputting polarized surface normal vectors as a priori knowledge to neural networks, so this paper only compares with previous U-net networks [14]. To further demonstrate the advantages of neural networks, this section of this paper also introduces algorithms from the literature [8,10], as shown in Figure 13, from left to right: the real normal vector, the algorithm proposed in this paper, the algorithm of Ba et al. [14], the algorithm of Smith et al. [8], and the algorithm of Mahmoud et al. [10].

It can be seen that Mahmoud et al. have a larger error when estimating normal vectors for objects with somewhat more complex textures, while Smith’s algorithm results in a large area of color distortion. In terms of texture, the algorithm in this paper obtains fuller estimated normal vectors than Ba et al., as seen, for example, in the neck part of the beast. This is because Ba et al. lack the Resnet structure in the encoder part, as shown in Figure 10; Monster under De_Resnet has a similar performance to that of the paper [14] in the red box. The Gan network, on the other hand, can guide the general trend of the fused image through the confrontation between the discriminator and the generator, which also promotes the higher accuracy of the result graph in this paper compared to Ba et al. Since MAE is a common metric used in normal vector estimation tasks, Table 1 mainly shows the comparison data between different algorithms for different target pixel values between 0 and 255. As shown in Table 1, the estimated normal vector accuracy obtained in this paper is improved by 1.5° compared to Ba et al. while there is a substantial improvement in accuracy compared to the other four algorithms. From Table 1, it can be seen that the model proposed in this paper has certain advantages over other algorithms in terms of estimating either smooth object surface normal vectors or simple texture normal vectors. Furthermore, this paper uses various algorithms for image fusion and normalizes the fused images to the 0–1 range. The target object in the image is then segmented using a mask, and the metrics MAE, MSE, SSIM, PSNR, CC, SCD, and AG are calculated between the estimated and true normal vectors of the target surface (as in Table 2). Among them, PSNR indicates the peak signal-to-noise ratio of an image and is usually used to measure the quality of image compression. However, PSNR is not the best choice for all image quality assessment problems because it only considers the differences between pixel values and not the differences perceived by the human eye, and thus may not match human perception in some cases. Therefore, in the evaluation of this paper, although the results of PSNR are poor, the other metrics are better, indicating that the proposed normal vector estimation algorithm works better.

In the reconstruction process, birds, monsters, and horses can represent more complex textures, while vases and Santa Claus can represent objects with smooth surfaces, so the results are chosen to be compared with those reconstructed in the literature [12,13], as shown in Figure 14 and Figure 15, from left to right: the reconstruction algorithm of this paper, the algorithms of Zhang [12], Yang et al. [13], Agrawal et al. [22], and Kovesi et al.’s [23] algorithm.

From the analysis of the reconstruction results, the estimated normal vector reconstructed surface texture of the object obtained in this paper is more prominent. Compared with the literature [12,13], the reconstruction in the literature [12,13] is too smooth, which leads to some gradient disappearance problems in the reconstruction results, such as the partial disappearance of the bird’s feathers, while the reconstructed result of this paper has a much clearer texture; the monster and horse reconstruction effect is similar to that of the bird. In the literature [22,23], the reconstructed surface of the bird appears distorted and the middle part of the surface of the vase is distorted, i.e., the reconstructed vase and bird results in this paper have an all-around advantage. In terms of relative implementation conditions, several other reconstruction algorithms require prior knowledge of depth map or point cloud data, which undoubtedly increases the workload of surface reconstruction. In contrast, the surface reconstruction framework proposed in this paper is easy to implement; only a set of single visual polarization images are required, and the reconstructed results have clearer textures. In summary, the algorithm proposed in this paper has some advanced significance.

## 5. Conclusions

The normal vector estimation task using the Gan network not only effectively avoids the problem of normal vector ambiguity, but also can make the estimated normal vector more full. Furthermore, the energy function constructed by the relationship between normal vector and gradient and height, and thus the height map idea, can avoid the problem of needing point cloud data in the reconstruction process, and the reconstructed surface texture is more obvious. The framework of the whole object surface reconstruction, which only requires a single set of single visual images, and the realization conditions are particularly simple.

A good interpolation will directly affect the normal vector estimation results in the reflective region correction process. Subsequently, the fusion interpolation algorithm is considered, which in turn supplements the reflective region. Secondly, the height map obtained by the preliminary constructed energy function has the problem of too much height. In this regard, constraint on the height map by adjacent points will be considered, so as to achieve higher reconstruction accuracy.

## Figures and Tables

**Figure 1 sensors-23-03638-f001:**
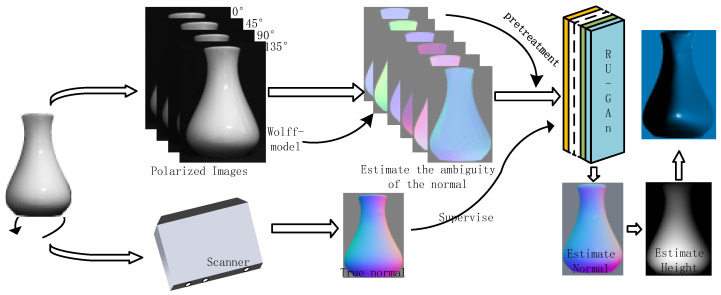
The proposed reconstruction framework.

**Figure 2 sensors-23-03638-f002:**
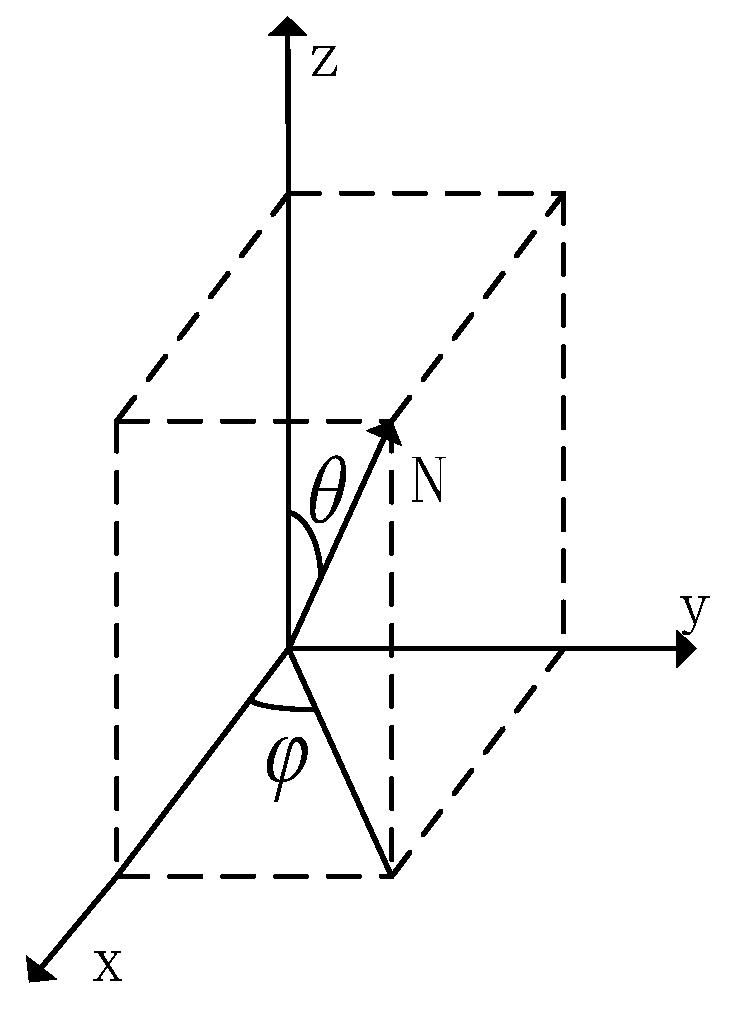
Polar coordinate representation of normal vector.

**Figure 3 sensors-23-03638-f003:**
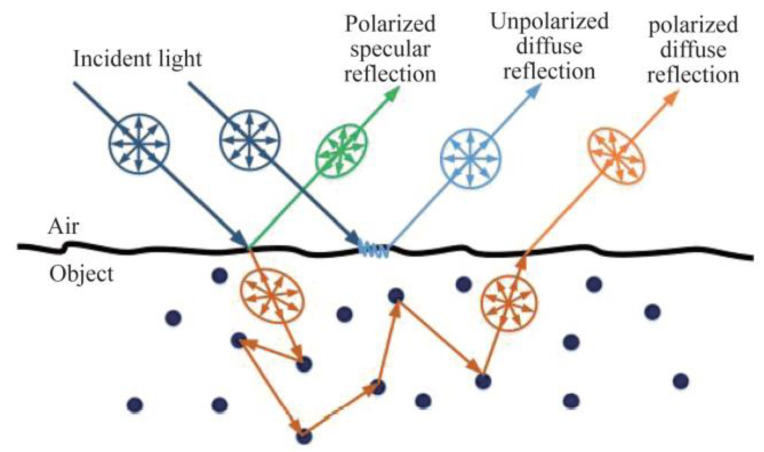
Schematic diagram of the emitted light from different material surfaces.

**Figure 4 sensors-23-03638-f004:**
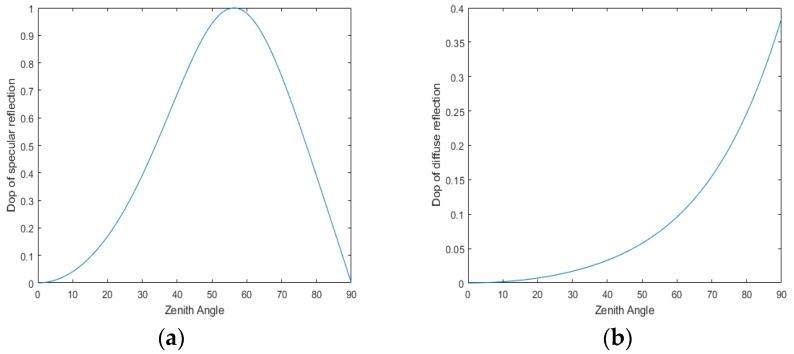
Relationship between zenith angle and physical model. (**a**) Specular reflection. (**b**) Diffuse reflection.

**Figure 5 sensors-23-03638-f005:**
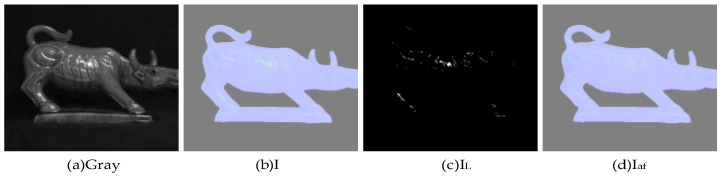
Schematic diagram of reflective correction.

**Figure 6 sensors-23-03638-f006:**
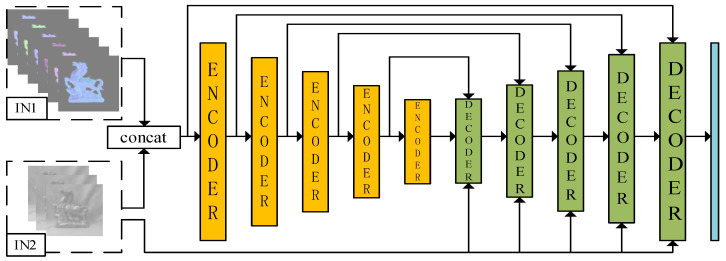
Generator network architecture diagram.

**Figure 7 sensors-23-03638-f007:**
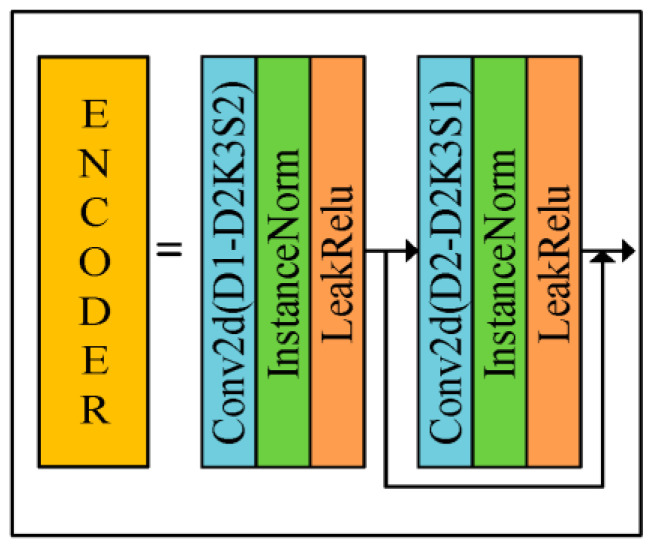
Encoder module.

**Figure 8 sensors-23-03638-f008:**
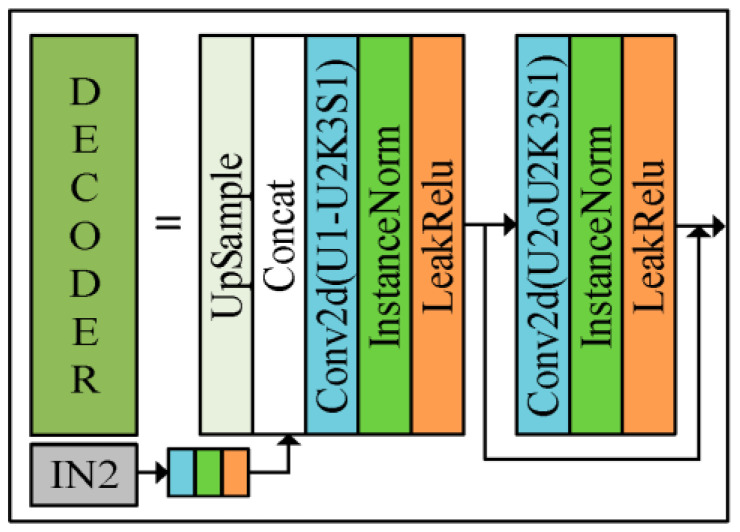
Decoder module.

**Figure 9 sensors-23-03638-f009:**
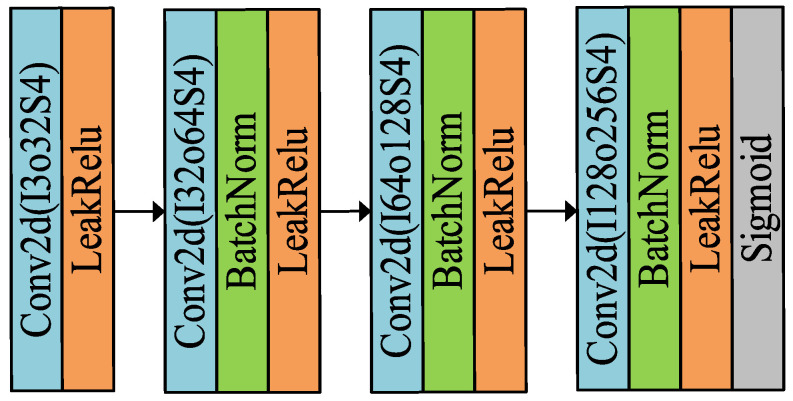
Discriminator architecture diagram.

**Figure 10 sensors-23-03638-f010:**
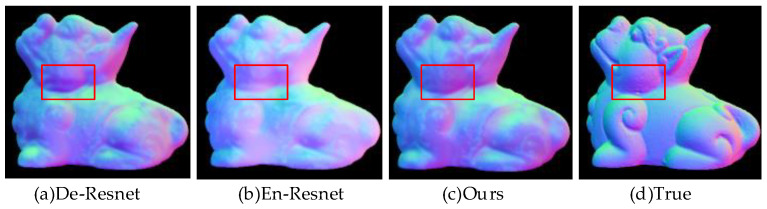
Resnet structure verification. The red boxes can be used to compare the differences in texture and color information brought by these three experiments.

**Figure 11 sensors-23-03638-f011:**
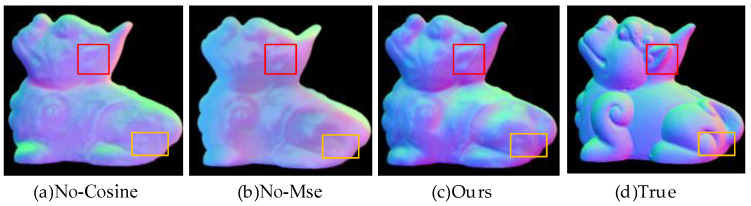
Validation of content loss. The red and yellow boxes can be used to compare the differences in texture and color information brought by these three experiments.

**Figure 12 sensors-23-03638-f012:**
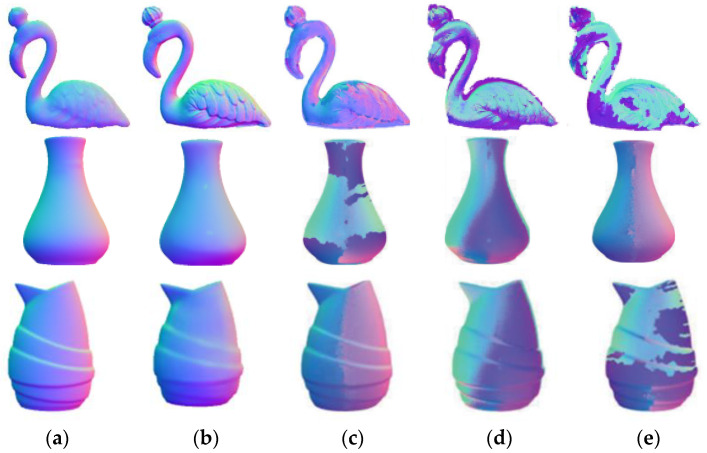
Comparison with traditional algorithms for estimating normal vectors. (**a**) True; (**b**) Ours; (**c**) References [12,13]; (**d**) Reference [9]; (**e**) Reference [10].

**Figure 13 sensors-23-03638-f013:**
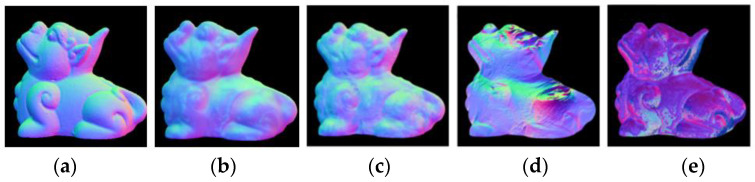
Comparison of neural network estimation normal vector algorithms. (**a**) True; (**b**) Ours; (**c**) Reference [14]; (**d**) Reference [8]; (**e**) Reference [10].

**Figure 14 sensors-23-03638-f014:**
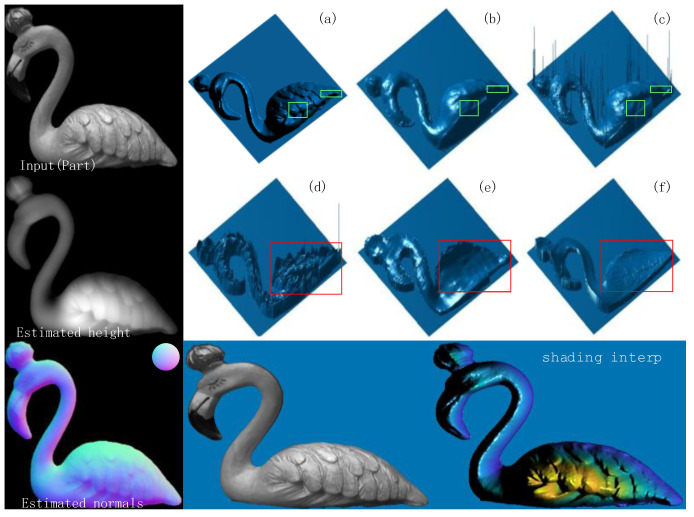
Bird reconstruction results. (**a**) Ours; (**b**) Reference [12]; (**c**) Multiple perspectives; (**d**) Reference [22]; (**e**) Reference [23]. (**f**) FC Algorithm. With the red and green boxes, the contrast between the reconstructed textures can be made.

**Figure 15 sensors-23-03638-f015:**
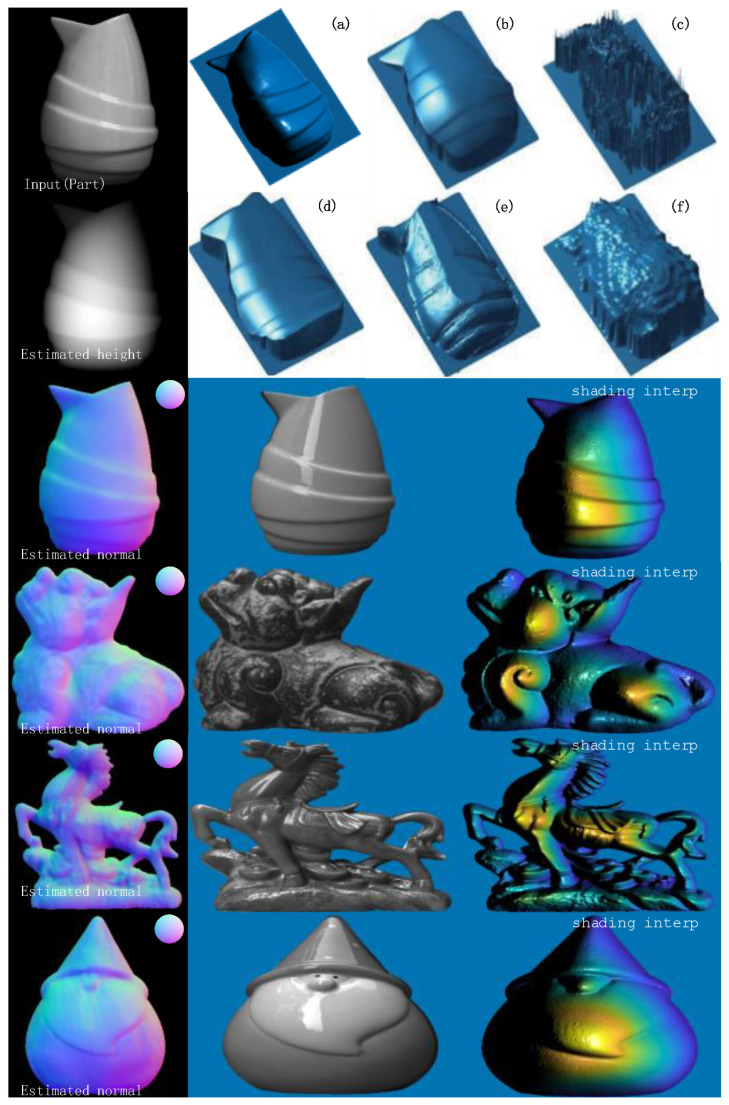
Sample of the rest of the reconstruction results. (**a**) Ours; (**b**) Reference [13]; (**c**) Multiple perspectives; (**d**) Reference [22]; (**e**) Reference [23]; (**f**) Rough depth.

**Table 1 sensors-23-03638-t001:** Comparison index of normal vector fusion algorithm.

MAE	Ours	Ba [14]	Zhang [12]	Yang [13]	Smith [8]	Mahm [9]	Miya [10]
Monster	20.05	21.55	30.21	28.21	45.25	70.16	51.45
Flamingo	18.75	20.19	25.86	24.92	34.54	41.25	42.42
Vase1	9.96	10.32	17.68	16.91	34.11	50.21	42.95
Vase2	11.80	12.30	16.68	16.82	40.21	44.70	50.00
Horse	21.83	22.27	33.48	31.72	42.76	47.79	47.95
Christmas	12.70	13.50	23.56	22.87	43.62	65.38	42.38

**Table 2 sensors-23-03638-t002:** Comparison of multiple metrics between different algorithms.

Indicators	Ours	Ba [14]	Zhang [12]	Smith [8]	Mahm [9]	Miya [10]
MAE	0.0643	0.0685	0.1152	0.1641	0.1931	0.1917
MSE	0.0121	0.0139	0.0237	0.0474	0.0569	0.0620
SSIM	0.9350	0.8379	0.7940	0.7450	0.7157	0.6562
PSNR	69.0172	69.3922	64.4152	61.5412	60.6321	60.4009
CC	0.9294	0.9193	0.8318	0.6441	0.5220	0.5432
SCD	0.0196	0.0224	0.0638	0.0662	0.0930	0.0941
AG	0.0536	0.0544	0.0926	0.1326	0.1591	0.1589

## Data Availability

https://drive.google.com/file/d/1EtjfMTfpanJotH92GFz300X_ZEmGXuqr/view (accessed on 11 February 2023).

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
