# Peer review of "Polarized Object Surface Reconstruction Algorithm Based on RU-GAN Network"

_sensors, 2023, doi:10.3390/s23073638_

Round 1

Reviewer 1 Report

This paper describes a proposed framework for surface reconstruction of objects from polarized images only, without the need for additional information such as shadows. The problem of normal ambiguity in the 3D reconstruction process is addressed by designing a U-shaped generative adversarial network (RU-Gan) based on jump compensation to fuse the six feasible solutions for surface normals. The use of jump compensation in the encoder and decoder parts of the network, and reconstruction of the content loss function, among other approaches, are proposed to address this issue. 

However, the reviewer believes the motivation in this paper is not very much clear. The reviewer does not understand what is shown in Line 81-82: At present, a better framework for object surface reconstruction from polarized images only has not been available. Moreover, the current challenges in this field need to be elaborated more. 

The novelty in this manuscript also needs to be improved to show the contribution compared to the SOTA.

Reviewer 2 Report

I have the following questions:

1. What is the innovation of the method? From the perspective of network architecture, the method does not integrate the target characteristics organically and fully. The author still needs to explain this aspect.

2. The experimental object seems to be a little simple, and the analysis of complex targets with multiple structures and errors should be clear, and what is the mathematical dependency between the parameterization requirements, methods and parameters of the corresponding network?

3. Single evaluation index: the author should provide multiple evaluation systems for comparison in terms of error and similarity.

Round 2

Reviewer 1 Report

The authors have addressed the issues regarding the reviews.

Reviewer 2 Report

All my concerns have been fixed